# Safety and Long-Term Immunogenicity of BNT162b2 Vaccine in Individuals with Down Syndrome

**DOI:** 10.3390/jcm11030694

**Published:** 2022-01-28

**Authors:** Diletta Valentini, Nicola Cotugno, Vittorio Scoppola, Chiara Di Camillo, Luna Colagrossi, Emma Concetta Manno, Carlo Federico Perno, Cristina Russo, Paolo Palma, Paolo Rossi, Alberto Villani

**Affiliations:** 1Pediatric Unit, Pediatric Emergency Department (DEA), Bambino Gesù Children’s Hospital, IRCCS, 00165 Rome, Italy; chiara.dicamillo@opbg.net (C.D.C.); alberto.villani@opbg.net (A.V.); 2Clinical & Research Unit of Clinical Immunology and Vaccinology, Academic Department of Pediatrics (DPUO), Bambino Gesù Children’s Hospital, IRCCS, 00165 Rome, Italy; nicola.cotugno@opbg.net (N.C.); emmaconcetta.manno@opbg.net (E.C.M.); paolo.palma@opbg.net (P.P.); 3Chair of Pediatrics, Department of Systems Medicine, University of Rome “Tor Vergata”, 00133 Rome, Italy; paolo.rossi@opbg.net; 4The School of Pediatrics, Tor Vergata University, 00133 Rome, Italy; vittorio.scoppola@opbg.net; 5Department of Laboratories, Bambino Gesù Children’s Hospital, 00165 Rome, Italy; luna.colagrossi@opbg.net (L.C.); carlofederico.perno@opbg.net (C.F.P.); 6Unit of Microbiology and Diagnostic Immunology, Department of Laboratories, Bambino Gesù Children’s Hospital, IRCCS, 00165 Rome, Italy; cristina.russo@opbg.net; 7Unit of Immune and Infectious Diseases, Academic Department of Pediatrics, Bambino Gesù Children’s Hospital, IRCCS, 00165 Rome, Italy

**Keywords:** down syndrome, COVID-19, mRNA vaccination, humoral response

## Abstract

We aimed to evaluate the safety and immunogenicity of the BNT162b2 vaccine in young people with Down syndrome (DS), and to compare their humoral immune response with those of the healthy controls (HC). Individuals with DS and HC received the BNT162b2 vaccine. Longitudinal blood samples were collected on the day of vaccination, twenty-one days after the first dose, seven days after the second dose, and six months after the first dose. Both the local and systemic adverse events reported by participants were mild. Pain at the injection site was the most reported local adverse event, while fever was the systemic adverse event. Humoral responses showed a significant increase of anti-S and anti-S trimeric antibody (Ab) levels after both doses of vaccine in both groups. In comparison with HC, Ab levels in individuals with DS were similar at T21, but significantly lower, both in terms anti-S and anti-S trimeric, at T28 (respectively *p* = 0.0003 and *p* = 0.0001). At T180 both groups showed a significant reduction of anti-S trimeric Ab levels compared to T28 (*p* = 0.0004 and *p* < 0.0001 for DS and HC, respectively). Individuals with DS exhibit a good humoral response to the BNT162b2 vaccine; however, similarly to in HC, the immune response wanes over time.

## 1. Introduction

Down syndrome (DS) is the most common chromosomal disorder and the leading genetic cause of intellectual disability in humans, which results from the triplication of chromosome 21. It is characterized by a higher rate of comorbidities, anatomical differences in the upper respiratory tract, and immune dysregulation, all of which increase the risk of severe illness due to infection by the novel severe acute respiratory syndrome coronavirus 2 (SARS-CoV-2) [1].

Recently, two reports were published by the Trisomy 21 Research Society, which were based on >1000 COVID-19 patients with DS and showed that children and adults with DS have significantly higher rates of lung complications and mortality than people without DS [2,3].

The immune defects associated with DS include mild to moderate T and B cell lymphopenia, a decrease of naive lymphocytes, impaired mitogen-induced T cell proliferation, and defects of neutrophil chemotaxis [4]. Furthermore, evidence of dysfunction of B lymphocytes in children with DS was demonstrated by Carsetti et al. [5], who found a significant decrease in switched memory B cells in peripheral blood taken from individuals with DS. Accordingly, several papers describe a suboptimal vaccine-induced immune response in people with DS [6,7,8].

To date, in Europe, four vaccines against SARS-CoV-2 have been approved by the EMA agency: two mRNA vaccines, and two attenuated adenovirus vector vaccines. Despite this, the vaccination campaign is proceeding in a non-homogeneous manner, with significant differences among countries. The identification of at-risk categories led to prior vaccine administration to vulnerable populations, especially the elderly and patients with comorbidities.

For the general population, the administration of a two-dose SARS-CoV-2 messenger ribonucleic acid (mRNA) vaccination was found to be safe and effective in immunocompetent individuals [9,10] and was approved for administration in December 2020. However, our group and others have shown that other cohorts of immune-compromised adolescents and young adults, such as solid organ transplanted patients [11,12] and those with primary immune deficiencies [13,14], have a suboptimal immunization with the routine vaccination schedule. In line with this, there is an urgent need to establish an effective immunization schedule against COVID-19 in individuals with DS previously known to have had a B cell dysfunction [5].

Given the lack of information on the safety and effectiveness of vaccines in general, and anti-SARS-CoV-2 in individuals with DS, in the present study, we first aimed to evaluate the safety of the SARS-CoV-2 mRNA vaccination in individuals with DS. Second, we measured SARS-CoV-2 specific antibodies (Ab) at baseline, twenty-one days after priming with the first dose, seven days after the second dose, and six months after the first dose. Finally, we correlated the humoral immune response of individuals with DS with those of the healthy controls (HC).

## 2. Materials and Methods

Forty individuals with DS were enrolled from 8 March to 26 October 2021, at the Down syndrome Centre of Bambino Gesù Children’s Hospital in Rome, Italy. Clinical characteristics are shown in Table 1. Out of the 40, two individuals with DS were excluded due to the presence SARS-CoV-2 anti-N antibodies at the initial blood draw, prior to vaccination. Thirty-eight patients were naïve to SARS-CoV-2 infection, as demonstrated by the absence of SARS-CoV-2 anti-N antibodies and no history of COVID-19 disease; they all received BNT162b2 vaccine, with a schedule of two doses of 30 mg, 21 days apart [15]. Longitudinal blood samples were collected the day of vaccination (T0), twenty-one days after the first dose (T21), seven days after the second dose (T28), and six months (T180) after the first dose. Health care workers, who received BNT162b2 vaccine, were used as HC. All participants with DS, assisted by caregivers, completed a questionnaire about potential adverse events and side effects following each dose of vaccine; each symptom was rated by the respondent as mild, moderate, severe, or absent. Local and systemic adverse events were defined: mild if reaction went away on its own and/or no doctor was needed; moderate if a doctor, clinic visit, or hospital admission were needed; and severe if the reaction was immediately life-threatening, or if emergency department, urgent care facility visit, or ICU admission were needed, or if reaction led to disability, permanent damage, or death of the patient. (https://www.who.int/vaccine_safety/initiative/tech_support/Part-3.pdf?ua=1; accessed on 4 January 2021). All procedures performed in the study were in accordance with the ethical standards of the institutional research committee and with the 1964 Helsinki declaration and its later amendments. A local ethical committee approved the study, and written informed consent was obtained from all participants or legal guardians (2409_OPBG_2021).

Venous blood was collected in ethylenediamine tetra-acetic acid tubes and processed within two hours. Plasma was isolated from blood and stored at −80 °C. Anti-SARS-CoV-2 IgG Ab titers were measured, as previously described [16] at each time point. We measured Ab against the S1-receptor-binding-domain (Roche, cut-off: 0.8 U/mL) and anti-S trimeric SARS-CoV-2 Ab (LIASION^®^ SARS-C0V-2 DiaSorin, cut-off: 13 AU/mL).

Statistical analyses were performed using GraphPad Prism 8 (GraphPad Software, Inc., San Diego, CA, USA). Statistical significance was set at *p* < 0.05 and all tests were two-tailed. All data were analyzed by D’Agostino–Pearson to assess the normality of the distributions; parametric and non-parametric tests were used for normally or non-normally distributed datasets, respectively. As indicated in the figure legends, paired and non-paired tests were used to assess differences between Ab load at the different time points, and between DS and HC, respectively.

## 3. Results

### 3.1. Local and Systemic Adverse Events after First and Second BNT162b2 Dose

Most participants with DS reported no local and systemic adverse events, and of those who reported events, they were rated as mild (Figure 1). Local adverse events were reported by 10 individuals with DS (25%) within seven days after the first dose; a similar proportion reported local reactions after the second dose (Table 2). The most reported local adverse event was pain at the site of injection for both the first and the second dose. Systemic adverse events were reported in five of the 40 (12.5%) individuals with DS after the first dose and in six of the 40 (15%) after the second dose. The most reported systemic reaction was fever after the second dose (four of the six reporting an event, Table 2). None required clinical care nor were hospitalized due to adverse reactions.

### 3.2. Humoral Responses after First and Second BNT162b2 Dose

At T0, 38 out of 40 individuals with DS were negative for anti-N, anti-S, and anti-S trimeric Ab, thus excluding previous immunization or infection with SARS-CoV-2. Humoral responses showed a significant increase of anti-S and anti-S trimeric Ab, both after the first (T21) and the second dose (T28) of BNT162b2 vaccine in both individuals with DS and HC (*p*-values all <0.0001; Figure 2A). Only one individual with DS showed no seroconversion after the first dose. Whereas no differences were found between the two groups at T21, a significantly lower SARS- CoV-2 Ab level was shown at T28 in individuals with DS compared to HC (*p* = 0.0003 for anti-S Ab levels and *p* = 0.0001 for anti-S trimeric Ab levels; Figure 2B). At T180, Ab levels in DS were similar to HC, but both groups had significantly reduced anti-S trimeric Ab levels compared to T28 (*p* = 0.0004 and *p* < 0.0001 for DS and HC respectively; Figure 2A). Comparing HC with DS, no difference in anti-S Ab levels was observed at T180; however, HC presented a significantly lower level of anti-S trimeric Ab compared to individuals with DS (*p* = 0.014; Figure 2B).

## 4. Discussion

In line with data from the general population [17], our data support a good safety profile for individuals with DS. No patients contracted a SARS-CoV-2 infection up to six months of observation after vaccination. Our results show the vaccine to be highly effective in this population, as measured by antibody levels, with only one individual with DS not seroconverting after the first dose, and compared to other vulnerable populations [11,12,13,14], which showed a suboptimal response to the routine vaccination schedule.

Although antibody levels were significantly lower in individuals with DS compared to HC at T28, no differences were found at T180. Indeed, the six-month serological test showed a similar titer between DS and HC groups in anti-S Ab levels and a slightly higher anti-S trimeric level in DS vs HC. One possible hypothesis explaining the similar waning between HC and DS is the older age of the HC. Indeed, the major impact of age on the maintenance of SARS-CoV2 specific immunity has been widely shown [18,19,20]. However, DS is a chronic condition, characterized by an accelerated immune aging, and an older aged control group can provide an important comparison, in terms of vaccine-induced immunity and memory maintenance.

We and others, previously described alterations in the immune response, characterized by a low frequency of naïve CD4+ T cells and by a reduced number of all B-cell populations in the peripheral blood and especially of switched memory B cells, also impacting on vaccine induced response [5,21,22]. Such perturbations may also underlie the reduced humoral response at T28 compared to HC. However, it is not yet known why this perturbation is compensated over time in individuals with DS, and additional analyses will be required to define the molecular mechanisms driving such delayed immunity upon mRNA SARS-CoV2 vaccination in persons with DS. Furthermore, we showed that in vitro switched memory B cells of individuals with DS have an increased ability to differentiate into antibody-forming cells in response to TLR9 signals. Thereafter, we showed that after primary vaccination against influenza, children with DS generate significantly less specifically switched memory B cells than their siblings [8]. Thus, individuals with DS respond well to vaccination with antibody secretion; however, they are less able to produce and maintain switched memory B cells. The identification of the mechanisms responsible for the low number of switched memory B cells is a prerequisite for the development of tailored vaccine protocols for the increasing populations with DS.

Considering this, individuals with DS should be prioritized for booster/third dose of mRNA vaccine, to improve their humoral response to vaccination.

Our study is the first showing that individuals with DS exhibit an initial good humoral response to the BNT162B2 vaccine, similarly to HC; even if it decreased in comparison with HC soon after the second dose. Our study has two primary limitations: a relatively small sample size, and the lack of age-matched controls. However, with regards to age differences, as individuals with Down syndrome may have accelerated immune aging [23], an older control group may be appropriate. Nevertheless, further studies on the immune cellular response to the BNT162b2 vaccine are needed to confirm our data and to identify the best immunization strategy against COVID-19 in this vulnerable population.

In conclusion, our findings confirm the good safety and immunogenicity profile of the BNT162b2 mRNA COVID-19 vaccine in individuals with DS and reinforce current national and international vaccine recommendations against COVID-19.

## Figures and Tables

**Figure 1 jcm-11-00694-f001:**
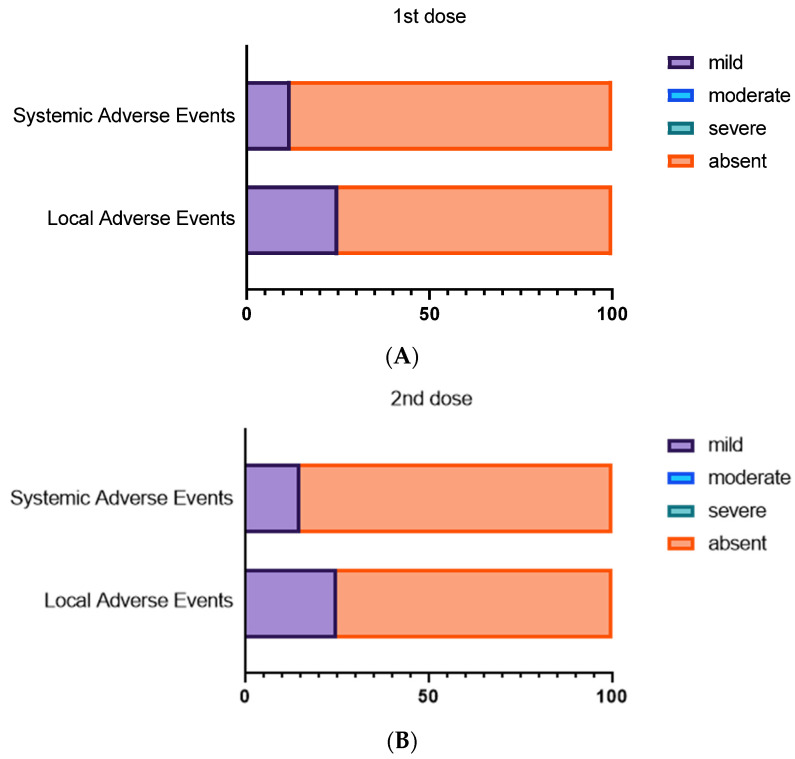
Local and systemic adverse events after first and second BNT162b2 dose. Contingency plots show safety profile following first (**A**) and second dose (**B**) of BNT162B2 mRNA COVID-19 vaccine in the DS cohort divided according to systemic or local adverse events.

**Figure 2 jcm-11-00694-f002:**
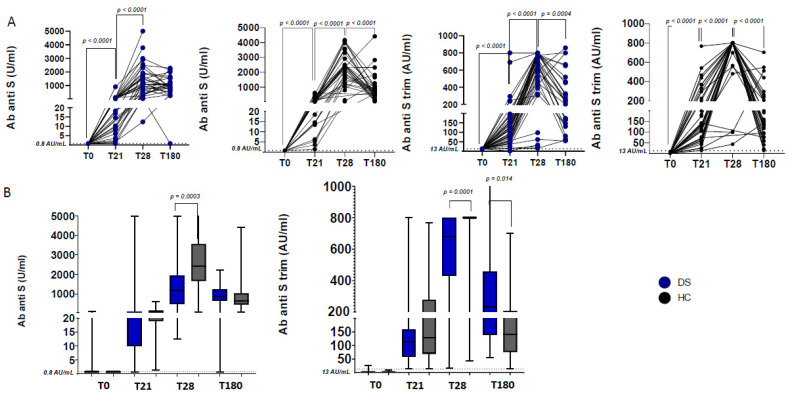
Antibody responses after first and second BNT162b2 dose. (**A**) Longitudinal analysis of anti-S and Ab anti-S trimeric Ab in HC (**black circles**) and DS (**blue circles**). Paired non-parametric *t*-test at adjacent time points within groups was performed to determine changes in longitudinal Ab levels. (**B**) Comparison of Ab levels among HC and DS. Differences between HC and DS levels were calculated for anti-S trimeric Ab, anti-S Ab at T0, T21, T28, and T180. Unpaired non-parametric *t*-tests were used for comparisons.

**Table 1 jcm-11-00694-t001:** Demographic and clinical characteristics.

Demographic and Clinical Characteristics	DS *n* = 40	HC *n* = 36	*p* Values
Females, *n* (%)	17 (42.5%)	23 (63.9%)	n.s.
Age, mean (±SD)	17.90 (±4.59)	47.4 (±12.2)	<0.0001
Ethnicity, *n* (%)	Black 1 (2.5%) East Asian 2 (5%) Latin American 4 (10%) White 33 (82.5%)	n.a.	n.a.
Type of trisomy, *n* (%)	Full/standard 37 (92.5%) Mosaic 0 (0%) Translocation 0 (0%) Don’t known 3 (7.5%)	n.a.	n.a.
Comorbidities, *n* (%)	Congenital heart disease 24 (60%) Recurrent respiratory infections 5 (12.5%) Thyroid disease 21 (52.5%) Celiac disease 3 (7.5%) Obstructive sleep apnea 15 (37.5%) Obesity 9 (2.5%) Hypertension 1 (2.5%) Chronic liver disease 3 (7.5%) Chronic lung disease 2 (5%) GERD 6 (15%) Allergies 2 (5%) Psychiatric disease 5 (12.5%)	n.a.	n.a.
Level of intellectual disability, *n* (%)	Mild 12 (30%) Moderate 15 (37.5%) Severe/profound 5 (12.5%) Don’t known 8 (20%)	n.a.	n.a.

**Table 2 jcm-11-00694-t002:** Adverse event description in the DS cohort.

	*n* = 40 1st Dose	*n* = 40 2nd Dose
Systemic adverse events	5 (12.5%)	6 (15%)
Fever	2/5	4/6
Wheezing	1/5	0
Muscle pain	1/5	1/6
Fatigue	1/5	2/6
Headache	1/5	1/6
Chills	1/5	0
Dizziness	1/5	0
Cough	1/5	1/6
Local adverse events	10 (25%)	10 (25%)
Pain at the site of injection	10/10	10/10
Redness at the site of injection	1/10	2/10

## Data Availability

The data presented in this study are available on request from the corresponding author. The data are not publicly available due to privacy.

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
