# Peer review of "Safety and Long-Term Immunogenicity of BNT162b2 Vaccine in Individuals with Down Syndrome"

_jcm, 2022, doi:10.3390/jcm11030694_

Round 1

Reviewer 1 Report

  1. The names of other marketed vaccines and the rationale of the study must be elaborated in the Introduction section.
  2.  I did not find Table 2 in the manuscript, which is an essential part of the results and discussion.
  3. A comparision of similar study with the current study must be provided in the discussion part along with future pespectives of the study.  

Author Response

We thank the Reviewer for the interesting revision of the manuscript. We have tried to improve the manuscript following the Reviewer's Comments. All changes/edits are highlighted in yellow in the revised text of the manuscript and a point-by-point reply to Reviewer’s concerns is enclosed below.

We hope the revised version of our manuscript could be suitable for publication.

Reviewer: 1

As suggested by reviewer1 we have improved the introduction, the results and the discussion as follows:
Comments to the Author1

1. the Introduction must be improved:

- The names of other marketed vaccines and the rationale of the study must be elaborated in the Introduction section.

Reply: We thank you for this correction; we have included other marketed vaccines and all relevant references on the introduction

2. the results can be improved:

- I did not find Table 2 in the manuscript, which is an essential part of the results and discussion.

Reply: We thank the referee for this criticism; we have added the table 2 on the results

3. the discussion must be improved

- A comparision of similar study with the current study must be provided in the discussion part along with future pespectives of the study.

Reply: we thank for this comment; we have described a comparison with other vulnerable populations (organ transplanted patients and those with primary immune deficiencies).

Reviewer 2 Report

The article is well written but there is some information missing.

  1. Please define in a table the symptoms that are classified as systemic and local adverse events, as well as the criteria to classify them as mild, moderate, and severe.

Please fix the resolution of figure 2, it is highly pixelated. Also, I would suggest changing the graph in figure 2B to a box and whisker plot.

Author Response

We thank the Reviewer for the interesting revision of manuscript. We have tried to improve the manuscript following the Reviewer's Comments. All changes/additions are highlighted in yellow in the revised text of manuscript and a point-by-point reply to Reviewer’s concerns is enclosed below.

We hope the revised version of our manuscript could be suitable for publication.

Reviewer: 2

Comments to the Author:

1. the Introduction must be improved

Reply: we thank you for this comment; we have included all relevant references on the introduction

2. the methods and the results must be improved:

-Please define in a table the symptoms that are classified as systemic and local adverse events, as well as the criteria to classify them as mild, moderate, and severe.

Reply: we thank for this comment; we have defined the criteria to classify the adverse events in the materials and methods section.

- Please fix the resolution of figure 2, it is highly pixelated. Also, I would suggest changing the graph in figure 2B to a box and whisker plot.

Reply: we thank for this criticism; we made the corrections.

Round 2

Reviewer 1 Report

Accept in present form